# The Identification of ECG Signals Using Wavelet Transform and WOA-PNN

**DOI:** 10.3390/s22124343

**Published:** 2022-06-08

**Authors:** Ning Li, Fuxing He, Wentao Ma, Ruotong Wang, Lin Jiang, Xiaoping Zhang

**Affiliations:** 1School of Electrical Engineering, Xi’an University of Technology, Xi’an 710048, China; 2180320030@stu.xaut.edu.cn (F.H.); mawt@xaut.edu.cn (W.M.); 2Key Laboratory of Control of Power Transmission and Conversion (SJTU), Ministry of Education, Shanghai 200240, China; 3Department of Electrical Engineering and Electronics, University of Liverpool, Liverpool L69 3GJ, UK; rw2016@liverpool.ac.uk (R.W.); l.jiang@liverpool.ac.uk (L.J.); 4Department of Electronic, Electrical, and Systems Engineering, School of Engineering, University of Birmingham, Birmingham B15 2TT, UK; x.p.zhang@bham.ac.uk

**Keywords:** electrocardiogram signal identification, wavelet transform, probabilistic neural network, mean impact value, whale optimization algorithm

## Abstract

Electrocardiogram (ECG) signal identification technology is rapidly replacing traditional fingerprint, face, iris and other recognition technologies, avoiding the vulnerability of traditional recognition technologies. This paper proposes an ECG signal identification method based on the wavelet transform algorithm and the probabilistic neural network by whale optimization algorithm (WOA-PNN). Firstly, Q, R and S waves are detected by wavelet transform, and the P and T waves are detected by local windowed wavelet transform. The characteristic values are constructed by the detected time points, and the ECG data dimension is smaller than that of the non-reference detection. Secondly, combined with the probabilistic neural network, the mean impact value algorithm is used to screen the characteristic values, the characteristic values with low influence are eliminated, and the input and complexity of the model are simplified. Finally, a WOA-PNN combined classification method is proposed to intelligently optimize the hyper parameters in the probabilistic neural network algorithm to improve the model accuracy. According to the simulation verification on three databases, ECG-ID, MIT-BIH Normal Sinus Rhythm and MIT-BIH Arrhythmia, the identification accuracy of a single ECG cycle is 96.97%, and the identification accuracy of three ECG cycles is 99.43%.

## 1. Introduction

With the development of information technology, biometric signal recognition technology has become increasingly important as a kind of information security system. raditional biometric signal recognition systems mainly use fingerprints, human faces, iris and other physiological characteristics for recognition [1,2,3]. Despitehe advantages of a higher recognition rate, faster recognition and higher measurability, these physiological characteristicslso have some disadvantages, such as being easy to copy and forge [4,5].

In recent years, the electrocardiogram(ECG) signal has proved to be effective for identification. Compared with the external physiological characteristics of organisms, the ECG signals can be measured only in a living body, so the ECG identification method is not easy to forge, which can improve the security of the access control system and ensure that the important information is not stolen. In addition, ECG signals have the characteristics of universality, uniqueness, stability and measurability [6]. Nowadays, wth the development of ECG data collection technology, portable ECG signal collection devices such as smartwatches have been designed in a highly convenient and intelligent manner. Therefore, identity recognition based on ECG signals has extensive applications [7].

The research of identity recognition based on ECG signals is mainly divided into two aspects: ECG signal detection and ECG signal identity recognition. Detection is to preprocess ECG signals to obtain easy-to-classify data; identification is to use classification algorithms to classify the detected data.

ECG signal detection is divided into benchmark detection and non-benchmark detection [8,9]. The benchmark detection method is to extract the characteristic points of the P wave, QRS wave and T wave from ECG signals [10,11,12] and then classify and recognize them according to such characteristics as time and amplitude. However, because the slight changes in the position of the detection point may lead to classification errors, the recognition accuracy is not high. The non-benchmark detection method is based on Fourier transform [13], empirical mode decomposition [14,15] and wavelet transform [16,17] to extract information from ECG signals without using characteristic points. However, due to a large amount of information, the recognition scale is small, and it takes a long time. Therefore, how to reduce characteristic information while improving accuracy has become the main issue of the research.

ECG identification methods include support vector machine (SVM) and back propagation (BP) neural network [18], which are not suitable for multi-target classification, and their accuracy is not high enough. There are also methods such as deep learning [19,20], convolutional neural networks [20,21], and some improved methods [22,23], which have high accuracy but require very high-performance computer equipment.

The ECG identity recognition block diagram proposed in this papers shown in Figure 1. First, combiningenchmark detection and non-benchmark detection methods, wavelet transform is used to extract Q, R and S (QRS) waves to obtain their time points; and local windowed wavelet transform is used to extract P and T waves to obtain their time pointsLocal windowed wavelet transform can avoid the low extraction accuracy caused by the phenomenon that the R peak is too large. Secondly, the probabilistic neural network (PNN) algorithm is used for ECG identification. The PNN multi-target classification algorithm has the advantages of being a simple process, fast convergence and high sample error tolerance [24,25]. Finally, the PNN algorithm is improved from the two aspects of accuracy and complexity. On the one hand, the mean impact value (MIV) algorithm [26,27] is used for variable selection, which simplifies the complexity of the algorithm and eliminates the characteristic values with large errors in the ECG detection and extraction process. On the other hand, therobabilistic neural network by whale optimization algorithm (WOA-PNN) is proposed, which uses WOA [28,29,30] to optimize the smoothing factor in PNN, improve the accuracy of the model classification and solve the problem that the smoothing factor of the PNN algorithm needs to be artificially given.

The contributions of this paper are as follows:The local windowed wavelet transform is used to extract P and T waves and obtain their time points, which can avoid the problem of a too-large R peak, which affects the extraction accuracy.The MIV algorithm is used to optimize the characteristic values of ECG identification in the PNN, eliminate the characteristic values with large errors in the detection or extraction process and simplify the algorithm complexity.The WOA-PNN algorithm is proposed to adaptively optimize the hyper parameters in the ECG identification model to improve the accuracy of the model.Experiments were performed on different ECG signal databases, including two normal ECG signal databases and one arrhythmia ECG database, to verify the robustness of the proposed method.

The rest of this papers arranged as follows: Section 2 introduces ECG characteristics detection based on wavelet transform. Section 3 introduces the ECG recognition using the WOA-PNN algorithm and performs variable selection on the ECG characteristics. Section 4 uses different ECG database simulations and verifies the effectiveness and robustness of the method by comparing and analyzing various methods. Finally, the results are discussed, and the conclusion is drawn in Section 5.

## 2. Ecg Characteristic Detection Based on Wavelet Transform Algorithm

This section mainly introduces the wavelet transform algorithm for QRS process detection [31,32], and the local windowed wavelet transform algorithm for detecting P and T wave processes proposed in this paper.

### 2.1. Qrs Wave Detection

The characteristics of the QRS wave of the ECG signal are relatively clear, which contain most of the characteristic information of the ECG signal waveform. Therefore, to obtain the characteristics of the ECG signal, QRS complex information needs to be extracted. In the QRS waveform, the R wave is the most recognizable waveform with the largest amplitude and the most obvious characteristics. The peak point of the R wave is generally composed of a four-layer discrete wavelet through a binary spline wavelet filter to obtain scales 1 (S1), 2 (S2), 3 (S3) and 4 (S4), respectively [31]. The research demonstrates that the R wave peak is the most different from the rest of the noise signal on S4. Therefore, this paper determines to locate the R wave peak on S4. Specific steps of detection are presented in the following.

Figure 2 shows a schematic diagram of R wave peak detection. First, it reads the ECG signal data. Secondly, the wavelet decomposition of scales of 1–4 was carried out, and the threshold was set on S4 to find the extreme value pair containing the R peak and its zero-crossing point. Then, the offset correction of the zero-crossing point was performed in the time domain; finally, the R wave peak point was found near the corrected position. After the above steps, the R peak position was accurate.

The research shows that peak values of Q and S waves are small, resulting in the absence of extreme points on S3 and S4, and the extreme points on S2 are not obvious enough. Therefore, this paper detects Q and S waves on S1. Three extreme points before the R peak value are approximately used as the starting point of the Q wave, and three extreme points after the R peak value are approximately used as the end point of the S wave.

### 2.2. P Wave and T Wave Detection

The detection of P and T waves can also be achieved by wavelet transform. However, the amplitude of R and P/T waves is too large, and the detection accuracy will be reduced when the wavelet scale transform is performed. Therefore, this paper proposes a locally windowed wavelet transform to detect P and T waves. The specific detection steps are as follows:

Step 1. Read ECG data.

Step 2. Perform QRS detection according to Section 2.1, and calculate the average period *T${}_{R}$* of the R peak.

Step 3. Set the window *W*${}_{P}$ and *W*${}_{T}$ of the wavelet transform, Q wave goes forward *W*${}_{P}$ sampling points for 1–4 scale wavelet transform, and S wave goes backward *W*${}_{T}$ sampling points for 1–4 scale wavelet transform.

Step 4. Find a value less than the given threshold on S4, which is the peak of P and T waves.

Step 5. If step 4 does not exist, *W*${}_{P}$ translates *a* × *n* to the left, *W*${}_{T}$ translates *a* × *n* to the right, repeat step 4, until the end of *W*${}_{P}$ + *a* × *n* >* T${}_{R}$*/2 or *W*${}_{T}$ + *a* × *n* > 2/3*T${}_{R}$*. where *a* is the translation amount of each time; *n* is translation times; *T${}_{R}$*/2 and 2/3*T${}_{R}$* are used to avoid exceeding the detection range.

Step 6. After finding the P and T wave peaks, find the maximum point on S4, which is approximately the starting point, and the ending point is approximately symmetrical to the starting point with respect to the peak point.

Figure 3 is a schematic diagram of P and T wave peak detection. Figure 3 is the electrocardiogram of a cycle, and the P and T waves are translated and windowed. Figure 3b,c are the P and T waves after multi-scale wavelet transformation, respectively. Only S3 and S4 are shown here. It can be seen that the minimum value of S4 corresponds to the peak point. The detection accuracy of the R peak point is already very high. Recognition based on the characteristic values of the R wave can only be used in smaller identification systems, so it is necessary to add the characteristics of P and T waves. However, current research on P and T waves are not yet mature. Therefore, the use of the above method will also cause problems, such as error detection and false detection, but the overall identification effect can be improved.

## 3. Ecg Identification Based on the WOA-PNN Algorithm

This section first introduces the theory of the PNN multi-target classification algorithm and applies it to ECG identification. Secondly, the MIV algorithm is used to simplify the algorithm input characteristic values, eliminating the low impact and wrong characteristic values. Finally, the smoothing factor in the WOA adaptive optimization PNN algorithm is proposed to improve the accuracy of ECG identification.

### 3.1. Introduction to PNN Algorithm

PNN is a feedback neural network based on the Bayes classification criterion and probability density function, which takes an exponential function as the antecedent of the activation function [24,25]. As a typical classifier, it is often used in pattern classification, fault prediction and other fields. Compared with the BP neural network used in the field of ECG signal identification, PNN is easy to train, with a fast convergence speed, strong scalability and strong ability for multi-target classification. The PNN network topology consists of the input layer, the pattern layer, the summation layer and the output layer, as shown in Figure 4.

The input layer transmits the characteristic parameters into the network. The number of input layers is the number of sample features. The pattern layer is connected with the input layer through the connection weight to calculate the matching degree between the characteristic vector and each mode in the training set, and its distance is substituted into the Gaussian function to obtain the output of the pattern layer. The number of neurons in the pattern layer is the number of input sample vectors. The summation layer connects the units of the pattern layer. The number of neurons in the summation layer is the number of categories of samples. The output layer is responsible for outputting the category with the highest score in the summation layer. The solution method of the PNN pattern layer is as follows:(1)$$\;\phi_{ij}\left(X\right)=\frac{1}{{\left(2\pi \right)}^{\frac{1}{2}}\delta^{d}}\exp[-\frac{\left(X-X_{ij}\right){\left(X-X_{ij}\right)}^{T}}{\delta^{2}}]$$
where *X* represents the input sample, δ represents the smoothing factor and *d* represents the number of sample attributes.

According to Equation (Equation 1), the sum is obtained, and the mean value is taken, then the solution method of the *i*-th mode is
(2)$$\;g_{i}\left(x\right)=\frac{1}{L}\sum_{j=1}^{L}\phi_{ij}\left(x\right)$$
where *L* represents the number of samples in the *i* mode.

Sum all modes and judge the result of the summation as follows:(3)$$\;y=\arg\max(g_{i})$$

Suppose the number of tests is *N*; $y_{i}$ is the actual value; ${\hat{y}}_{i}$ is the PNN output value.
(4)$$\;E\left(y\right)=\frac{1}{N}\sum_{i=1}^{N}{\left(y_{i}-{\hat{y}}_{i}\right)}^{2}$$

Continuously optimize *E*(*y*), adjust parameters such as smoothing factor, and obtain the required stable network model.

### 3.2. Miv Algorithm Characteristic Values Screening

Suppose that the number of input variables of the prediction model shown in Equation (Equation 1) is *p*. Let the *p* variables form an independent variable vector, and make *m* times observations to get the independent variable space of $X=[x_{1},x_{2},\cdots ,x_{m}]$. Accordingly, the dependent variable corresponding to each sample point can be written as $Y=[y_{1},y_{2},\cdots ,y_{m}]$. Take ***X*** composed of independent variable vectors of *m* samples as the input and the corresponding vector ***Y*** as the output to form a training sample set {***X***, ***Y***} to train the neural network, and save the trained neural network. Then, the independent variable space for the original training is transformed as follows: the respective variables are added and subtracted α (%), respectively, on the basis of the original value to obtain the following 2*p* (*i* = 1, 2, …, *p*) new independent variable spaces.
(5)$$X_{i}^{\left(1\right)}=\left[\begin{array}{cccc} x_{11} & x_{12} & \cdots & x_{1m}\\ x_{21} & x_{22} & \cdots & x_{2m}\\ \vdots & \vdots & \cdots & \vdots \\ x_{i1}\left(1+\alpha \right) & x_{i2}\left(1+\alpha \right) & \cdots & x_{im}\left(1+\alpha \right)\\ \vdots & \vdots & \cdots & \vdots \\ x_{p1} & x_{p2} & \cdots & x_{pm} \end{array}\right]$$
(6)$$X_{i}^{\left(2\right)}=\left[\begin{array}{cccc} x_{11} & x_{12} & \cdots & x_{1m}\\ x_{21} & x_{22} & \cdots & x_{2m}\\ \vdots & \vdots & \cdots & \vdots \\ x_{i1}\left(1-\alpha \right) & x_{i2}\left(1-\alpha \right) & \cdots & x_{im}\left(1-\alpha \right)\\ \vdots & \vdots & \cdots & \vdots \\ x_{p1} & x_{p2} & \cdots & x_{pm} \end{array}\right]$$

The constructed new independent variable space is used as the input of the neural network model in turn, and after the network output, the 2*p* outputs corresponding to the *i*-th (*i* = 1, 2 …, *p*) input variable index in the change sample points are obtained when the index of input variables changes.
(7)$$Y_{i}^{\left(1\right)}=\left[\begin{array}{cccc} y_{i1}^{\left(1\right)} & y_{i2}^{\left(1\right)} & \cdots & y_{im}^{\left(1\right)} \end{array}\right]$$
(8)$$Y_{i}^{\left(2\right)}=\left[\begin{array}{cccc} y_{i1}^{\left(2\right)} & y_{i2}^{\left(2\right)} & \cdots & y_{im}^{\left(2\right)} \end{array}\right]$$

The difference operation is performed on the vectors in Equations (Equation 7) and (Equation 8) to obtain the influence change value vector of the output value after the change of the *i*-th input variable index in each sample point, expressed as $I_{V.i}=Y_{i}^{\left(1\right)}-Y_{i}^{\left(2\right)}$, so as to obtain the average influence value of the output value of *m* times when the index of the *i*-th input variable changes, expressed as follows:(9)$$I_{MIV.i}=\sum_{j=1}^{m}\frac{I_{V.i}\left(j\right)}{m}i=1,2,\cdots ,p$$
where $I_{\mathrm{MIV}.i}$ is the average influence value of the *i*-th input variable index on the output result in the change sample. The sign $I_{\mathrm{MIV}.i}$ indicates the direction in which the independent variable is related to the dependent variable, and the absolute value represents the relative importance of the independent variable’s influence on the dependent variable.

### 3.3. WOA Parameter Adaptive Optimization

#### 3.3.1. Introduction of WOA

Inspired by this special predation behavior, Seyedali Mirjalili et al. proposed a new swarm intelligence optimization algorithm—the Whale Optimization Algorithm (WOA), in 2016 [28,29,33]. WOA simulates the predatory behavior of whales in the ocean; optimizing the search through the process of whales surrounding prey and using bubbles to attack the prey. Consistent with classical particle swarm optimization, ant colony algorithm and artificial bee colony algorithm, the WOA is, in essence, a process of statistical optimization. The WOA has been widely concerned by many scholars for its advantages of simple operation, few parameters and excellent performance, and has been applied to solve different practical problems, such as electric vehicle charging optimization, solar cell parameter optimization and photovoltaic cell parameter optimization. In this paper, the WOA is applied to ECG signal identification. When solving the multidimensional nonlinear equation of the PNN algorithm, smoothing factor parameter δ needs to be optimized to obtain the minimum error of the model and avoid the tedious manual parameter setting process.

The WOA imitates the foraging behavior of whales to find optimal solutions to parameters, including three position update methods: surrounding prey, rotating search and random search.

(1) Surrounding prey

The whale shares the information of the target prey, and then it approaches the other whales, which are closest to the prey in the current group, and gradually shrinks the encirclement of the whole whale group to surround the prey. The whale position update formula is
(10)$$\;\;\;\;\;\;\;\;\left\{\begin{array}{c} X\left(t+1\right)=X^{*}\left(t\right)-A\times D\\ D=\left|C\times X^{*}\left(t\right)-X\left(t\right)\right| \end{array}\right.$$
where *t* represents the number of iterative searches, *X* represents the position of the whale, *X** represents the optimal global position and *A* and *C* represent the prey, which is expressed as
(11)$$\;\;\;\;\;\;\;\;\left\{\begin{array}{c} A=2a\times r_{1}-a\\ C=2r_{2}\\ a=2-2t/T_{\max} \end{array}\right.$$
where $r_{1}$ and $r_{2}$ are uniformly distributed random numbers between [0, 1], *a* represents the convergence factor, which decreases linearly from 2 to 0, and $T_{\max}$ represents the maximum number of iterations.

(2) Rotating search

Whales search for the prey in a spiral upward and slowly approach the target. The expression of the spiral search is
(12)$$\;\;\;\;\;\;\;\;\left\{\begin{array}{c} X\left(t+1\right)=X^{*}\left(t\right)+D\times e^{bl}\cos\left(2\pi l\right)\\ D=\left|C\times X^{*}\left(t\right)-X\left(t\right)\right| \end{array}\right.$$
where *b* is a constant, which can change the shape of the spiral, and *l* is a uniformly distributed random number between [−1, 1].

When a whale is searching for the prey in a spiral, it also shrinks its encirclement, so in order to simulate this behavior, it is necessary to simultaneously surround the prey and search in the spiral. The updated formula is
(13)$$\;\;X\left(t+1\right)=\left\{\begin{array}{c} X^{*}\left(t\right)-A\times D\;\;\;\;\;\;\;\;\;\;\;\;\;\;\;\;\;\;\;\;p<0.5\\ X^{*}\left(t\right)+D\times e^{bl}\cos\left(2\pi l\right)\;\;\;\;\;p\geq 0.5 \end{array}\right.$$
where *p* is a random number uniformly distributed between [0, 1].

(3) Random search

In order to improve the global search ability of whales, let the whales search for prey with a certain degree of randomness and increase the search range of whales.

When the coefficient |A| < 1, it means that the whale is in the constricted encircling circle and the rotating search method is selected. When the coefficient |A| ≥ 1, it means that the whale is outside the constricted encircling circle, and the random search method is selected. The random search update formula is as follows:(14)$$\;\;\;\;\;\;\;\;X\left(t+1\right)=X_{rand}\left(t\right)-A\times \left|C\times X_{rand}\left(t\right)-X\left(t\right)\right|$$
where $X_{rand}$ is a random whale position.

#### 3.3.2. WOA-PNN Algorithm

This paper proposes the WOA-PNN algorithm to solve ECG signal identification. First, the PNN algorithm is used to classify and train the ECG signal. PNN has the advantages of simple structure, concise training and strong nonlinear classification ability. However, the smoothing factor δ in PNN significantly affects the classification accuracy. Selecting the value of δ is a complicated process and requires intelligent optimization. Therefore, this paper proposes to use the WOA to intelligently optimize the smoothing factor δ, set the fitness function to measure the advantages and disadvantages of the individual’s spatial position and use the whale foraging strategy to continuously update the individual whale position until the optimal whale spatial position is obtained, which is the optimal smoothing factor δ for the PNN algorithm.

As shown in Figure 5, first, we input the ECG characteristic information extracted by the wavelet transform algorithm, set the initial smoothing factor δ of the PNN algorithm and create a PNN network. Then, we calculate the error of the PNN network and use it as the fitness value of the WOA. When the fitness does not meet the demand, use the WOA to update the smoothing factor δ. When the number of iterations is less than the given number, recreate the PNN network model. Until the fitness value or the times of iteration meets the requirements, the optimal ECG identification model is finally obtained.

## 4. Simulation Experiment

### 4.1. Experimental Data

The original ECG data used in this paper are from PhysioNet [34]. PhysioNet is a free resource supported by the National Academy of Medical Sciences (NIGMS) and the National Institute of Biomedical Imaging and Bioengineering (NIBIB) that provides physiologic signal libraries and processing tools for researchers. In this paper, the ECG-ID database [35], MIT-BIH Normal Sinus Rhythm database [36] and MIT-BIH Arrhythmia database [37] are selected for experimental verification. The ECG-ID database has 90 ECG signals from subjects with normal ECG signals. As the main database to verify the identification algorithm in this paper, the MIT-BIH normal sinus rhythm database contains 18 ECG signals from subjects with normal ECG signals, which plays an auxiliary role in verification. The MIT-BIH arrhythmia database has the ECG signals of 48 arrhythmia subjects, which is used to compare the simulation with other studies, verify the effectiveness of the method and verify the general applicability of the method.

### 4.2. Results of Qrs Wave, P Wave and T Wave Detection

This study used 10 ECG signal cycles from each subject, and a total of 1560 ECG signal cycles were compared. After multiple instances of manual professional observation and the intelligent identification of positions, the detection accuracy results of the intelligent method are shown in Table 1 below. There were some differences in the results. The R wave had the most obvious characteristics, and its detection result was 100%; the detection results of other waves were also above 80%. The data of the MIT-BIH arrhythmia database were the ECG signals of subjects with arrhythmia, so the detection accuracy of this database was relatively low.

Figure 6 shows the results of the QRS wave detection part. Obviously, the detection of the R peak position was correct, the detection of the Q wave position was also correct, and the detection of the S wave position had very few errors. Figure 7 shows the detection results of the P and T waves. Compared with the T wave, the characteristics of the P wave were more obvious, so the detection accuracy of the position of the P wave was higher than that of the T wave. Although very few errors may occur in the above detection, the extracted positioning points could still be used as features for identification, which improved the accuracy of the ECG signal identification.

### 4.3. Ecg Identification Simulation Results of WOA-PNN Algorithm

This section first constructs the characteristic values based on the detection results of the previous section and uses the MIV algorithm to screen characteristic values with high influence. Secondly, based on the MIT-BIH arrhythmia database, it is compared with other studies to verify the feasibility of this method. Finally, in order to improve the accuracy, ECG signals are expanded from one cycle to three cycles, and the method is verified in different databases.

#### 4.3.1. Results of Characteristic Values Screening of the Miv Algorithm

Based on the results of QRS, P and T wave detection, 22 characteristic values were selected, including 16 distance characteristic values: R-R, R-Q, R-S, R-P, R-T, R-P${}_{begin}$, R-P${}_{end}$, R-T${}_{begin}$, R-T${}_{end}$, Q-P, Q-P${}_{begin}$, S-T, S-T${}_{end}$, P-T, P${}_{begin}$-P${}_{end}$, T${}_{begin}$-T${}_{end}$ and 6 amplitude characteristic values: R-Q, R-S, Q-P, S-T, P${}_{begin}$-P, T${}_{begin}$-T.

Table 2 shows the influence degree of the PNN characteristic values based on the MIV algorithm. The influence degree was arranged in descending order. The detection accuracy of the R wave was the highest, so the R-R characteristic values were the most accurate, and its influence degree was also the highest. The influence degrees of the correlation characteristic values of P and T waves were greater than those of Q and S waves. It could be concluded that the detection of P and T waves is necessary. The time of Q and S waves was relatively short, and the extraction results were not obvious. The influence degrees of all amplitude characteristic values were 0. The amplitude characteristics were obtained on the basis of the distance characteristics, so when the distance characteristic deviated, the amplitude characteristic deviation would be larger.

Figure 8 shows ECG identification of the PNN algorithm with different numbers of characteristic values. The characteristic values were added in descending order of MIV. It could be seen that the identification accuracy of the first 6 characteristic values reached more than 90%; after the 13th characteristic value was added, the identification accuracy reached the limit value, and the subsequent characteristic values had little effect on the classification accuracy. In addition, the 13th characteristic value point corresponded to the MIV value of 0.1, so this article used all the characteristic values with MIV values greater than 0.1 for ECG identification. Excluding nine characteristic values simplified the complexity of the algorithm by 40.91%.

#### 4.3.2. Single Ecg Cycle Identification Result Contrast

On the basis of the above experiment, the characteristic values with a MIV greater than 0.1 were used as the input variables for the PNN, and the different identities were numbered from 0 as the output label of PNN. Using 70% of the data as the training data and 30% of the data as the test data and compared to the other methods [33,38,39,40,41,42], a summary is made, as shown in Table 3.

The feature extraction and recognition method of ECG signal recognition based on the PNN network proposed in this paper has relatively high recognition accuracy. Compared with other methods, the absolute value of the accuracy is improved by 1.07% (MIT-BIH Arrhythmia) and 3.35% (ECG-ID database). There are few studies using the MIT-BIH Normal Sinus Rhythm database, so no comparative analysis is given here.

In order to improve the recognition accuracy, this paper sets the population size to 10 and iteration time to 100 to optimize the smoothing factor δ in the WOA-PNN. The iterative process of the WOA-PNN is shown in Figure 9. After 33 iterations, the ECG-ID database’s recognition accuracy was improved from 95.65% to 97.16%, the absolute accuracy increased by 1.51%, the recognition error rate decreased from 4.35% to 2.48%, and errors decreased by 43.67%. After 22 iterations, the recognition accuracy of the MIT-BIH Arrhythmia database was improved from 94.48% to 95.48%, the absolute accuracy increased by 1%, and the recognition error rate decreased from 5.52% to 4.52%, with a 19.93% reduction in error. The final smoothing factor δ was 5.6801. In summary, the weighted average accuracy of the WOA-PNN for ECG signal recognition of the three databases is 96.97%.

#### 4.3.3. Three Ecg Cycle Identification Results

Single ECG identification can be errant, so three ECG identification can be used to be more accurate, just to be on the safe side. Table 4 shows the ECG identification results of three ECG cycles under different databases. It could be seen that the identification accuracy of the three databases is above 98%. The WOA-PNN algorithm in this paper is superior to the traditional PNN algorithm, with a weighted average identification accuracy of 99.43%.

## 5. Conclusions

ECG signal biometric technology is rapidly replacing traditional fingerprint, face, iris and other recognition technologies, avoiding the vulnerability of traditional recognition technologies. This article proposes an ECG signal identification method based on the wavelet transform algorithm and WOA-PNN algorithm. Firstly, the wavelet transform was used to detect QRS waves and the local windowed wavelet transform was used to detect P and T waves, and the characteristic values were constructed according to the detection time point to reduce the dimension of the ECG data. Secondly, combined with the PNN, the MIV algorithm was used to screen the characteristic values, and the characteristic values with low impact were eliminated, simplifying the input and complexity of the model. Finally, the WOA-PNN combined classification method was proposed to intelligently optimize the hyper parameters in the PNN algorithm and improve the accuracy of the model. According to the experimental analysis, the accuracy of identification of a single ECG cycle was 96.97%, and the accuracy of identification of three ECG cycles was 99.43%.

The next step of this study is to further optimize the classification algorithm to reduce the impact of increasing categories. In addition, the follow-up research will introduce more databases, such as the PTB database, to validate the method.

## Figures and Tables

**Figure 1 sensors-22-04343-f001:**
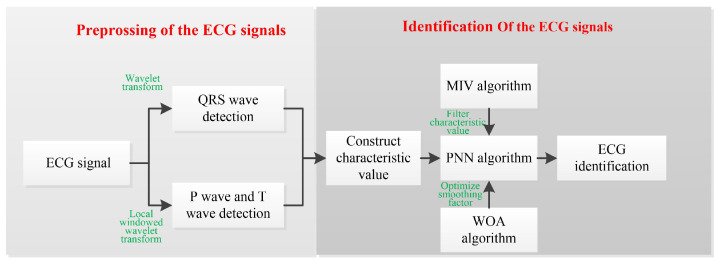
The ECG identification block diagram.

**Figure 2 sensors-22-04343-f002:**
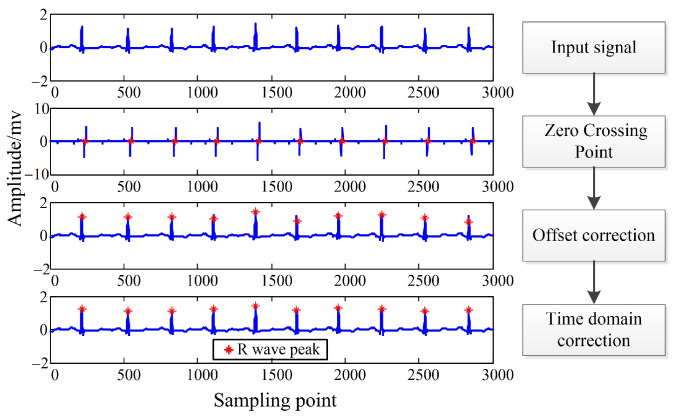
Schematic diagram of R wave peak detection.

**Figure 3 sensors-22-04343-f003:**
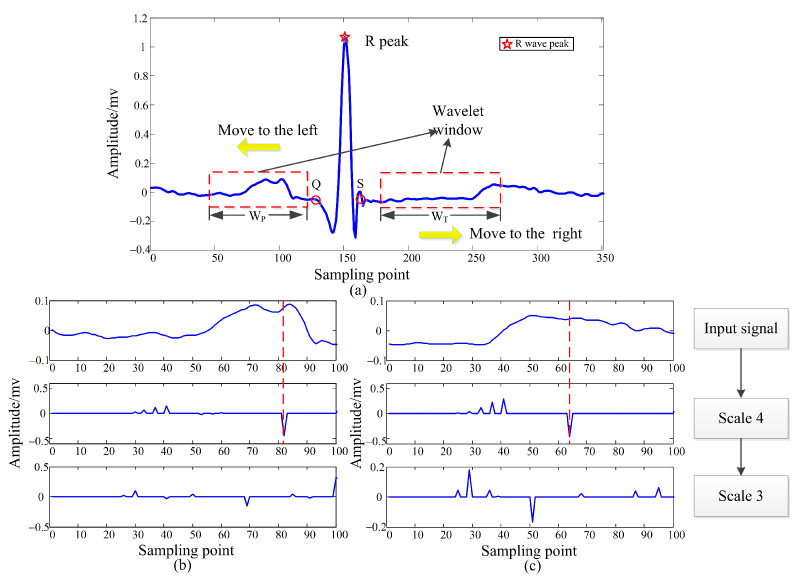
Schematic diagram of P and T wave peak detection. (**a**) Windowed wavelet transform of P and T waves; (**b**) P wave peak detection; (**c**) T wave peak detection.

**Figure 4 sensors-22-04343-f004:**
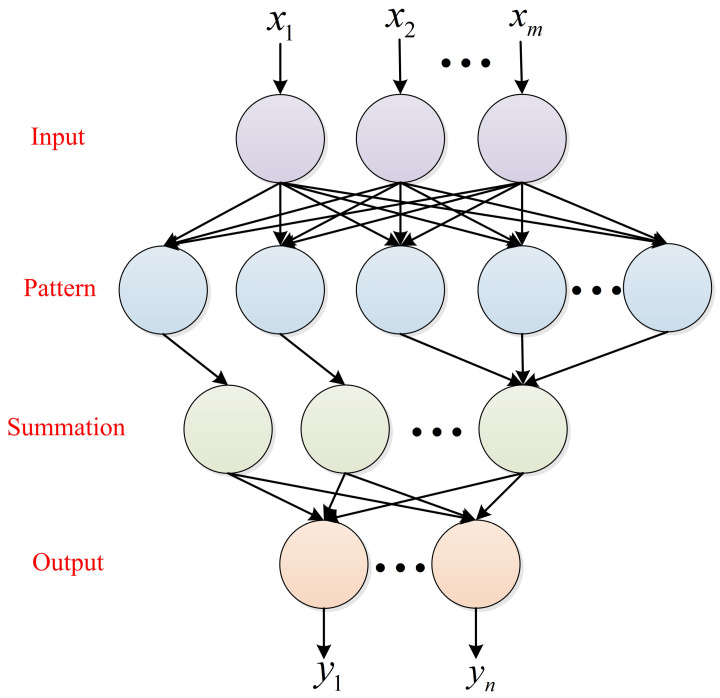
Basic structure of probabilistic neural network.

**Figure 5 sensors-22-04343-f005:**
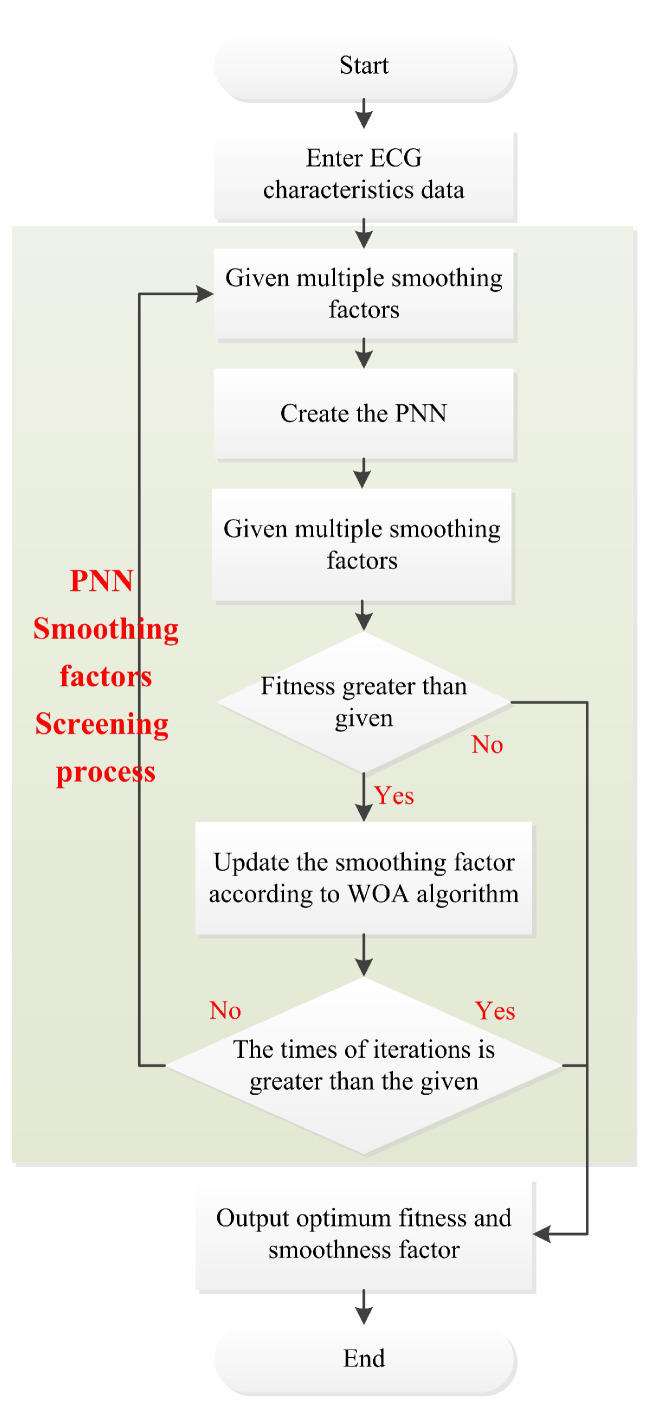
Flow chart of ECG signal identification based on the WOA-PNN algorithm.

**Figure 6 sensors-22-04343-f006:**
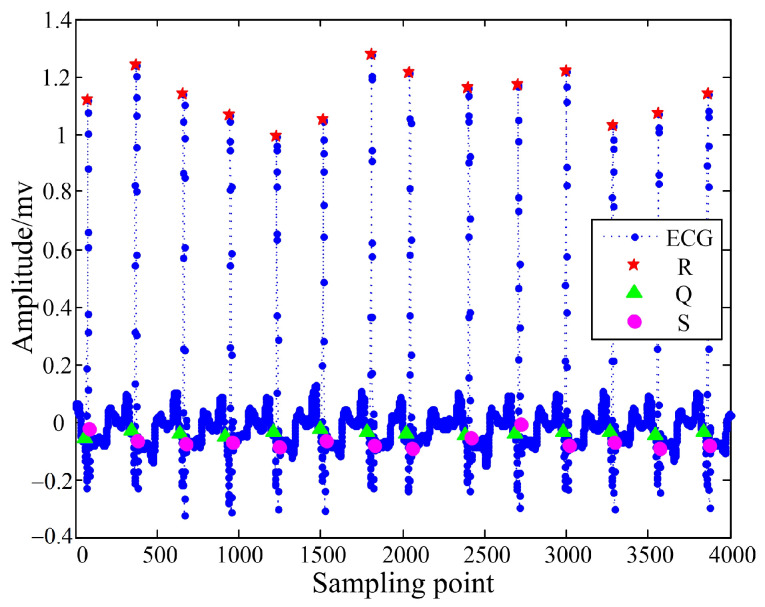
QRS wave detection results.

**Figure 7 sensors-22-04343-f007:**
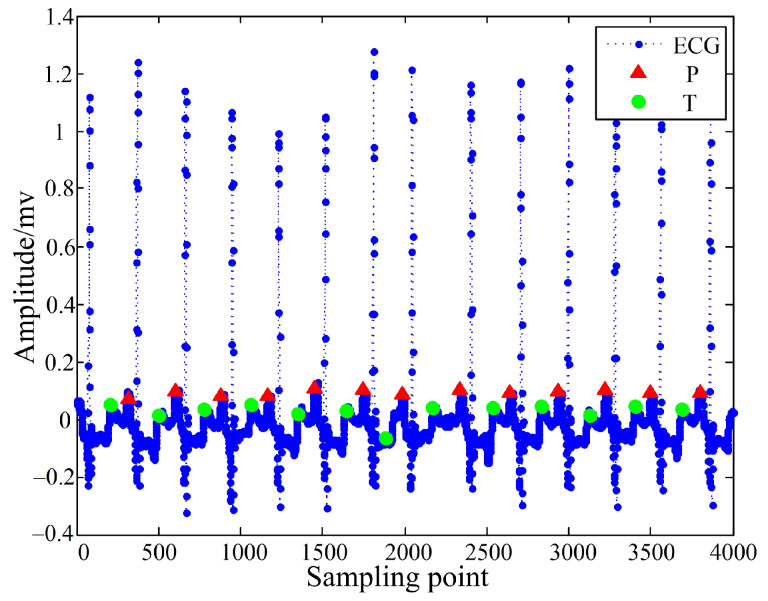
P wave and T wave detection results.

**Figure 8 sensors-22-04343-f008:**
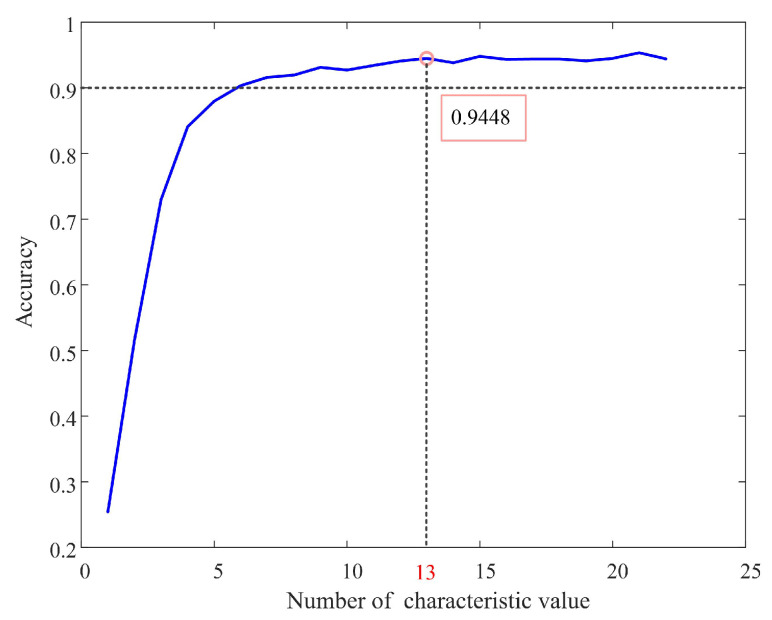
ECG identification of the PNN algorithm with different numbers of characteristic values. (The MIV was added from large to small).

**Figure 9 sensors-22-04343-f009:**
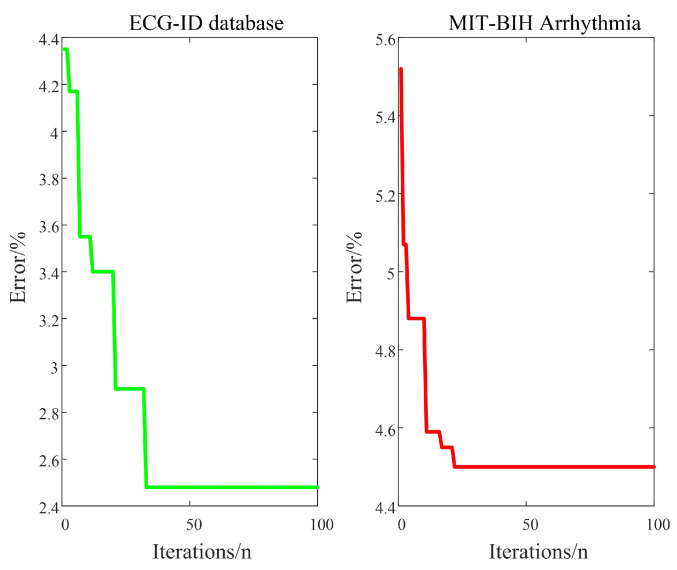
WOA-PNN algorithm iterative process diagram.

**Table 1 sensors-22-04343-t001:** QRS Wave, P Wave and T Wave detection results in different databases (%).

	Q	R	S	P	T
ECG-ID	93.24	100	91.96	89.24	87.52
MIT-BIH normal	89.57	100	87.08	84.28	82.93
MIT-BIH arrhythmia	82.47	100	81.03	80.80	78.83
Weighted average	89.50	100	88.03	86.07	84.32

**Table 2 sensors-22-04343-t002:** Characteristic value influence degree based on the MIV algorithm.

Distance	R-R	T${}_{\mathrm{begin}}$-T${}_{\mathrm{end}}$	R-T${}_{\mathrm{end}}$	R-P${}_{\mathrm{begin}}$	S-T${}_{\mathrm{end}}$	R-P
MIV	1.0862	0.6744	0.570	0.4713	0.3176	0.2987
Distance	Q-P${}_{begin}$	R-P${}_{end}$	P-T	Q-P	S-T	R-T
MIV	0.2449	0.2138	0.1724	0.07955	0.04933	0.02933
Distance	R-T${}_{begin}$	P${}_{begin}$-P${}_{end}$	R-Q	R-S		
MIV	0.01	0.00644	0.00222	0.00022		
Amplitude	R-Q	R-S	Q-P	S-T	P${}_{begin}$-P	T${}_{begin}$-T
MIV	0	0	0	0	0	0

**Table 3 sensors-22-04343-t003:** Comparison of ECG identification accuracy between PNN and the traditional Arrhythmia database.

Database	Method	Accuracy (%)
ECG-ID	WOA-PNN	97.16
	PNN	95.65
	Softmax [41]	92.3
	SFFS KNN [40]	91.26
	Random Forest [39]	83.9
	KNN [38]	83.2
MIT-BIH Arrhythmia	WOA-PNN	95.48
	PNN	94.48
	SVM [43]	93.41
	Decision tree [43]	92.68
	Random Forest [42]	92.68
	Bayes [42]	90.24
	Logistic [42]	83.54
	SVC [42]	83.52

**Table 4 sensors-22-04343-t004:** Identification of multiple ECG cycles (%).

	ECG-ID	MIT-BIH Normal	MIT-BIH Arrhythmia	Weighted Average
PNN	99.33	99.76	98.08	99.00
WOA-PNN	99.79	100	98.54	99.43

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
