# Peer review of "The Identification of ECG Signals Using Wavelet Transform and WOA-PNN"

_sensors, 2022, doi:10.3390/s22124343_

Round 1

Reviewer 1 Report

First of all, allow me to congratulate the authors for the interest aroused by the proposal presented. Very likely, the analysis and characterization of almost all biological signals will be applied in a short time for a better and safer identification of individuals in an increasingly digitalized society.

On the other hand, the article is well structured and correctly introduced, although its publication requires a very intense effort on the part of the authors to improve their writing. Many consecutive sentences appear unconnected with each other, looking more like a sequence of enumerations than an article. An example of this case can be found in the first 3 sentences of the introduction. It is necessary that the authors make an effort to generate texts with connectors that will help to the understanding of the document.

As for the methods presented and used in this study, important improvements are required. An example of this can be found in the case of WOA. In this particular case, the authors allude to the way whales behave, for illustrative purposes, but they do not clearly link this fact to the case presented. The authors must therefore consider that these types of tools are not always positively valued in scientific articles, but if they are, it should serve to very effectively illustrate the parallelism with the case study, and describe so.

The WOA, MIV, and PNN methods require a greater effort by authors for their correct description. In particular, it would be necessary to incorporate more bibliographic references that illustrate the use of these techniques together with this larger description.

The conclusions are too limited and the authors should incorporate additional efforts to discuss the results and add references in the literature that show the improvement found of this paper.

Reviewer 2 Report

The paper targets the problem of biometric identification using ECG. The authors have proposed some new approaches for the pre-processing of the ECG signal and for the task of classification/identification itself. Their method is tested on 3 commonly used databases for that application task, and compared to the performance of other methods for the same problem. While their approach achieves incremental improvement over the competition, there are several aspects in the paper, which can be improved. 

Major concerns:

  • The performance of the proposed approach is compared with other state-of-the-art methods only on one of the databases (MIT-BIH Arrhythmia), although experiments with the proposed method are performed on three databases. This can be seen as a serious lack in the experimental evaluation. Is there a reason for doing so?
  • The authors compare the performance of their methodology on 3 databases, one of which (EGC-ID) is a standard in the task in the ECG biometric identification task. The other one is the PTB database, which is not included in the analysis in the paper. Was there a reason for it to not be included? It would be beneficial to include it, since a lot of other studies on the same topic are using it.
  • The questions of uniqueness and permanence over time of the ECG markers are a burning issue when using ECG for biometric identification. Concerning uniqueness, it needs to be confirmed that ECG from one person is significantly different from the ECG of all other people. Permanence over time, on the other hand, requires that a person’s ECG does not change over some period of time, depending on the use-case. Please comment on the uniqueness and permanence of your method and the databases used.

Minor remarks:

  • The databases on which the experiments have been performed are not identified in the abstract. It is beneficial to have this information already in the abstract to better inform the reader (and for indexing).
  • Please avoid using abbreviations in the abstract (especially if they are not properly introduced, like PNN).

Round 2

Reviewer 1 Report

Please, next time it would be appreciated if a document was incorporated where the reviewers' suggestions are answered. Additionally, the document with multiple corrections in PDF does not facilitate the reviewer's reading and revision. Please, the publisher and the authors must be aware of the difficulty of reading a document with multiple colors and strikethroughs from previous versions. It is more appropriate to have one of the following options: (i) a single document where the additions are indicated in another color but not the deleted texts if it is done in PDF, including a separate text indicating compliance or not with the suggestions of the reviewers. (ii) a word document with changes control that allows the reviewer to see the final document for reading without difficulties.

Certainly this is one of the corrections where it has been more complex to re-read the document.

Reviewer 2 Report

The authors have made effort to appropriately address all the issues raised. I am still a little disappointed about not including the PTB database in the analyses. It would have made the experimental evaluation complete.